



# Attribution of aerosol particle number size distributions
# to major sources using a 11-year-long urban dataset

**Máté Vörösmarty[1], Philip K. Hopke[2,3], and Imre Salma[4]**

[1] Hevesy György Ph.D. School of Chemistry, Eötvös Loránd University, Budapest, Hungary

[2] Department of Public Health Sciences, University of Rochester School of Medicine and Dentistry, Rochester, NY, USA

[3] Institute for a Sustainable Environment, Clarkson University, Potsdam, NY, USA

[4] Institute of Chemistry, Eötvös Loránd University, Budapest, Hungary

**Correspondence:** Imre Salma (salma.imre@ttk.elte.hu) and Máté Vörösmarty (vmate6@student.elte.hu)

**Abstract.** Source apportionment was performed using size segregated particle number concentrations (PNCs) in 27 size channels over the diameter range of 6–1000 nm augmented by air pollutants all with a time resolution of 1 h in the urban background of Budapest for 11 full years in separate seasons. The input dataset was corrected for the effect of the local meteorology by dispersion normalization using the ventilation coefficient defined as the planetary boundary mixing layer height multiplied by the wind speed. Both the uncorrected and dispersion-corrected datasets were evaluated using positive matrix factorization. Six source types including nucleation, two road vehicle emission sources separated into a semi-volatile fraction and a solid core fraction, diffuse urban source, secondary inorganic aerosol (SIA), and ozone-associated particles were identified, characterised, and quantified. The ventilation correction substantially modified the input concentrations, while the differences in the corrected-to-uncorrected ratios for the contributions remained within 5 %. The overall mean relative contribution of the road traffic emission sources was 60 %, and did not show considerable seasonal variability. Nucleation was responsible for 20 % of the PNC annually as a lower estimate. It exhibited a compound character consisting of photochemically induced nucleation and traffic-related nucleation. The former process occurs on regional or urban spatial scales around noon, whereas the latter process happens when the gas-phase vapours in the vehicle exhaust cool, and the resulted supersaturated vapours nucleate outside the source. Its relative contributions were maximal in spring (somewhat smaller in summer and autumn) and minimal in winter. The contributions from the SIA and the urban diffuse source types were approximately 10 % in spring, summer, and 12−15 % in autumn and winter, respectively. The $O_3$-associated secondary aerosol made up the smallest (6 %) portion of particles on an annual basis. Directionality variations investigated by conditional bivariate probability function analysis were used to locate the likely source areas, and showed considerable spatial variations in the source origin.



## 1 Introduction and objectives

Particulate matter (PM) plays a vital role in the urban air quality worldwide. It is often quantified by the
mass of particles, which is established as a key or criteria air pollutant (EU EEA, 2023; US EPA, 2023).
Coarse- and accumulation-mode particles make up most PM mass, whereas the mass contribution of the
ultrafine (UF) particles (with $d < 100$ nm) is negligible (e.g., Salma et al., 2002). Despite that UF particles
make up > 80 % of total particle numbers in cities (Trechera et al., 2023). At relatively low PM mass and
high UF particle concentrations, it is the particle number that represents the potential danger to human
health better than the PM mass. Although there is less information on the role of UF particles in health
effects, there are toxicological (Oberdörster et al., 2005; HEI Review Panel, 2013), clinical (Chalupa et
al., 2004) and epidemiologic (Kreyling et al., 2006; Wang et al., 2019a) studies, which suggest that these
particles can cause adverse health effects. Inhalation of very small insoluble particles can particularly lead
to excess health risk relative to coarse or fine particles of the similar chemical composition (Oberdörster
et al., 2005; HEI Review Panel, 2013). This is caused by the vast number of the deposited particles in the
respiratory system, their large total surface area and small size (Braakhuis et al., 2014; Salma et al., 2015;
Riediker et al., 2019). The World Health Organization identified the UF particles as a potential risk factor
for humans (WHO, 2021).
Particle number size distributions (PNSD) can vary considerably over space and time. Formation and
atmospheric transformation processes basically contribute to this process. Apart from the vicinity of
intensive sources of UF particles, the PNSDs change rates become much slower. Under these balanced
conditions, the PNSDs can be separated into such size modes that are associated with source types or
aggregate sources (Hopke et al. 2022 and references therein). The PNSDs usually consist of Aitken and
accumulation modes. In addition, nucleation mode appears for constrained time intervals. Aitken-mode
particles are usually emitted into the air and can contain largely variable portions of semi-volatile
components condensed on solid (mostly soot) core (Morawska et al., 2008; Harrison et al., 2019; Rönkkö
and Timonen, 2019). Accumulation-mode particles ordinarily result from transformation processes such
as condensation growth, physical and chemical ageing or water activation processes of Aitken-mode or
nucleated particles. The nucleation mode can be associated with new aerosol particle formation (NPF)
and growth events (Kulmala et al., 2003).
Primary pollutants (including particle number concentrations and size distributions) can also be affected
by meteorological processes such as atmospheric mixing and transport due to their dispersion (dilution or
enrichment). The dispersion is often governed by solar radiation through planetary boundary mixing layer





height (MLH), wind or precipitation (Andronanche, 2004; Kumar et al., 2011). These conditions can
substantially affect both larger, geographically closed areas such as orographic basins and smaller
territories such as cities or valleys (Leahey, 1972; Salma et al., 2020). The dispersion of primary particles
is essentially related to the available air volume in which they are mixed (Holzworth, 1967; Ashrati et al.,
2009). In cities, this volume is determined by the MLH and WS in the first order approximation. It is
noted that meteorological variables may affect secondary pollutants and particles in a more complex and
separate way with respect to the primary pollutants and particles.

The spatial and temporal diversity and dynamics of the formation and transformation processes, and of
meteorological conditions are reflected in the PNSDs as far as both their integrated concentration and
shape are concerned (Li et al., 2023). Thus, size distributions can be used for identifying and quantifying
various source types. These basically differ from the sources dominating the PM mass. The particle
number concentrations are nonconservative compared to the PM mass. Attribution of PNSDs to different
source types and their quantification are desirable and essential since many basic properties, atmospheric
behaviour of particles as well as their health, environmental and climate effects depend on their number
(and not their mass) concentration (e.g., Ibald-Mulli et al., 2002; Meng et al., 2013; Corsini et al., 2019).
Source apportionment can also yield valuable knowledge for creating air quality regulatory strategies for
particle numbers or their source specific exposure metrics. Therefore, there is recently a considerable and
increasing scientific interest in source apportionment studies on PNSDs (Beddows et al., 2019; Dai et al.,
2021; Hopke et al., 2022; Teinilä et al., 2022; Conte et al., 2023; Crova et al., 2024; Rowell et al., 2024).
However, studies based on multiple-year-long data are still scarce (de Jesus et al, 2020).

Source apportionments can be achieved by multivariate modelling (Hopke, 1991). Positive matrix
factorisation (PMF; Paatero and Tapper, 1993, 1994) is one of the most widely used, well established and
efficient technique for this (Hopke, 2016; Hopke et al., 2020). The PMF modelling was successfully
applied to mass concentrations of aerosol constituents and gases (e.g., Viana et al., 2008; US EPA, 2014;
Hopke et al., 2020). The main differences between the PMF deployed on particle number size distribution
data with respect to that on mass concentrations include different approaches in handling zero data and
values below the detection limit, and in estimating the observation uncertainties (Ogulei et al., 2007).

To study the phenomenon of the urban atmospheric NPF and growth in Budapest, PNSDs in the diameter
range of 6–1000 nm, meteorological properties and air pollutants were measured for 11 full measurement
years. They belong to the longest critically evaluated urban datasets of this kind in the world. Utilising





this readily available dataset for source apportionment by PMF method offers different and
comprehensive insights into the sources of particle numbers. Such long-term observations are particularly
valuable as they can statistically reveal information which were hidden in the noise on shorter time scales
(Kulmala et al., 2023). The main objectives of this study are 1) to present and discuss the results and
experience gained from the source apportionment of PNSDs by applying the PMF method for separate
seasons in Budapest; 2) to quantify the effect and importance of the dispersion correction; 3) to interpret
the main sources and their spatial distributions; and 4) to determine the relative contributions from the
sources. Our conclusions can also contribute to the general understanding of the source, transformation
and transport processes of particle numbers in cities and to developing novel air quality regulatory policy
for them.
**2 Methods**
**2.1 Experimental part and data treatment**
The measurements were performed at two urban sites in Budapest. Most of them were conducted at the
Budapest platform for Aerosol Research and Training (BpART) Laboratory (47°28'29.9" N, 19°3'44.6"
E; 115 m above mean sea level, m.s.l.) of the Eötvös Loránd University (Salma et al., 2016a). The
measurement site is located 85 m from the River Danube, which flows through the city centre. The
location represents a typical urban background site due to its geographical and meteorological conditions.
The other measurement site was in a wooden area of the Konkoly Astronomical Observatory (47°30'00"
N, 18°57'47" E; 478 m above m.s.l.) at the NW border of the city. Since the prevailing wind direction in
the area is NW, the latter site represents the near-city background.

The PNSDs were measured using a flow-switching-type differential mobility particle sizer (DMPS)
system, which operates in the electrical mobility diameter range from 6 to 1000 nm in the dry state of
particles (relative humidity, RH < 30 %) separating the particles into 27 size channels with a time
resolution of $\tau = 8$ min (Salma et al., 2011, 2016b, 2021). The nominal diameters of the 27 channels are

123   6.0, 7.3, 8.9, 10.8, 13.2, 16.0, 19.5, 23.7, 28.9, 35.2, 42.9, 52.1, 63.4, 77.2, 93.9, 114, 139, 169, 206, 250,

304, 371, 451, 550, 670, 816, and 994 nm. This list facilitates the exact interpretation of the factor profiles
in Figs. 2a–4a and S5a–S7a. The concentrations of NO, $NO_x/NO_2$, CO, $O_3$, $SO_2$, $PM_{10}$ mass were acquired
from the closest measurement stations of the National Air Quality Network located 4.5 km from the urban
background site and 6.9 km from the near-city background site in the upwind prevailing direction (Salma
et al., 2020). The time resolution of these measurements was 1 h. Air temperature ($T$), RH, wind speed
(WS), wind direction (WD) and global radiation were measured at the BpART Laboratory and above the





rooftop level of the building complex (at a height of 45 m above the nearest street). The wind data above
the rooftop level were utilised in the present study and were recorded by standardized sensors (WAA15A
and WAV15A, both Vaisala, Finland) with τ = 10 min. Mixing layer height data (τ = 1 h) were extracted
from the Copernicus Climate Change Service (ERA5 Family datasets, ECMWF reanalysis; Hersbach et
al., 2023).

The data were expressed in local time (UTC+1 or daylight-saving time UTC+2). This was chosen since
the activities of the inhabitants greatly influence the atmospheric concentrations and size distributions in
cities (Mikkonen et al., 2020). Hourly mean particle number size distributions were derived from the
experimental data to reduce their fluctuations and the number of the missing data. Atmospheric
concentrations in each size channel and of the total particle number concentrations ($N_{6-1000}$) were
calculated and further evaluated. The investigated time interval involved 11 full measurement years
(Table S1). The data from the two urban sites were evaluated together. The residuals and the goodness of
the fits in the PMF modelling did not indicate significant differences between the respective factor profiles
in the urban background and near-city background. Additionally, this multi-site approach is expected to
improve the efficiency of the source apportionment (Pandolfi et al., 2010; Dai et al., 2020; Harni et al.,
2023). The median $N_{6-1000}$ and atmospheric concentrations of pollutants over the measurement years are
also summarised in Table S1. The overall dataset was finally split into separate subsets for meteorological
seasons (March, April, May as spring, June, July, August as summer, September, October, November as
autumn and December, January, February as winter) to fulfil one of the basic requirements of the PMF
method on the consistency of the source profile over the time interval considered (Zhou et al., 2004;
Ogulei et al., 2007). The missing concentration values in the input dataset were replaced by the medians
with 3-times the measurement uncertainty of the seasonal dataset. The data coverage for the input data
was typically > 85 %. The total number of observations for the PNSDs are shown in Fig. S2. The seasonal
means and standard deviations (SDs) of the meteorological properties are summarised in Table S2.
**2.2 Source apportionment modelling**
The source apportionment was performed using the PMF method with the equation solver Multilinear
Engine 2 (ME-2) as described by Hopke et al. (2023). The method decomposes the input dataset into a
factor (source) profile matrix and a factor contribution matrix with a user-specified factor number based
on the covariances between the variables. The PMF iteratively optimizes the objective parameter $Q$, which
is calculated on the individual residuals ($e$) and the uncertainties ($s$) for the observation $i$ and variable $j$:
$$Q = \sum_{i=1}^{m} \sum_{j=1}^{n} \left( \frac{e_{ij}}{s_{ij}} \right)^2 , \tag{1}$$





where $m$ and $n$ are the maximum number of observations and variables, respectively. $Q_{true}$ was calculated
with all data points, whereas $Q_{robust}$ was determined excluding the poorly fitted data points (i.e. when their
uncertainty-scaled residuals were greater than 4). The uncertainties of the particle number concentrations
in a size channel $j$ were estimating as (Ogulei et al., 2007):
$$\sigma_{ij} = (A * \alpha) * \left(N_{ij} + \overline{N}_j\right), \tag{2}$$

$$s_{ij} = \sigma_{ij} + C_3 * N_{ij}, \tag{3}$$

where $\sigma$ is the estimated individual measurement uncertainty for an observation, $N$ represents the observed
concentration, $\overline{N}$ is the arithmetic mean of the observed concentrations in the respective variable, $\alpha$ is
constant (of 0.01), which value is fine-tuned by $A$ for particle number concentrations, $s$ is the overall
uncertainty matrix, and $C_3$ is constant (0.01 for size channels, 0.2 for $N_{6-1000}$ and 0.15 for air pollutants),
which is also tuned around these nominal values. These selections and relationships are widely accepted
in the PNSD source apportionment studies. The addition of the pollutants is beneficial for the PMF as the
new quantities provide insights into the sources or atmospheric processes that produce the measured size
distributions, and reduce the rotational ambiguity of the model by complementing the edge points
(Paatero, 1999; Hopke, 2016). Specifying too low uncertainties relative to the true error level results in
overweighting those datapoints, while larger uncertainties yields downweighting (Hopke, 2020).
Moderate downweighting exerts less sensitive effect on the modelling results than overweighting, and the
overdetermined uncertainties can also obscure the concentration data.

Dispersion of the atmospheric concentrations due to the changes of meteorological conditions can result
in additional covariance as well. This effect can be corrected for by dispersion normalization of the input
dataset with the ventilation coefficient (VC; Ashrati et al., 2009). In this approach, the available air
volume for the atmospheric dispersion is proportional to the product of the $MLH_i$ and the vectorial mean
of the wind speed ($u_i$) for the observation $i$:
$$VC_i = MLH_i * u_i. \tag{4}$$

The concentration data ($C_i$) were multiplied by the ratio of the corresponding $VC_i$ and its seasonal mean
value $\overline{VC}$ (called ventilation coefficient ratio):
$$C_{Vi} = C_i * \frac{VC_i}{\overline{VC}}. \tag{5}$$

After completing the PMF analysis on the corrected dataset, the derived source contributions were divided
by the respective VC ratios to obtain the real contributions. The source apportionment modelling was
performed independently both on the uncorrected and corrected concentrations. The results derived from





the uncorrected dataset (i.e. $C_i$ concentrations) are referred as uncorrected (or traditional) PMF data, while
those obtained from the corrected dataset ($C_{Vi}$ concentrations) are denoted as dispersion-corrected (DC-)
PMF data.

The PMF solutions were explored in 50 runs with different configurations for each dataset. The factor
count was changed between four to twelve; the uncertainty parameters were modified from 0.01 to 0.05
for ($\alpha \times A$), and between 0.01 and 0.1 for $C_3$. Increased uncertainty settings were adopted for the smallest
(< 10 nm) and the largest (> 800 nm) size channels since their uncertainties were proven to be larger
(Wiedensohler et al., 2012), and for the pollutants since they were set as weak variables. The final solution
was reached through a trial-and-error approach. Additional error estimations were run using bootstrap
and displacement analyses. From the analysis point of view, the best solution (approved later as the final
solution) was chosen to meet the criteria that the convergence is achieved in the robust manner; its $Q_{true}$
and $Q_{robust}$ diagnostic values are among the lowest values; the scaled residuals are distributed preferably
normally between –3 and +3; and that the goodness of the fit (expressed by the coefficient of
determination, $R^2$) for the strong variables are reasonable (typically > 0.85). From the interpretation
aspect, the main requirements were that the solution is physically interpretable based on the size profiles;
is acceptable as far as directional probability function plots are concerned, and shows sensible diel
patterns, weekly and annual tendencies.

Spatial variations of the source intensities and other properties were derived by conditional bivariate
probability function (*polarPlot()*) of the 'openair' package (Carlsaw and Ropkins, 2012; Uria-Tellaexte
and Carlsaw, 2014). The method utilizes WS and WD data to create plots of directionality. The plots
derived from the uncorrected and corrected PMF modelling were compared using the *polarDiff()* function
of the package. Further statistical evaluations and presentations were accomplished by a laboratory-
developed application AeroSoLutions2 in conjunction with the Accord.NET Framework (Souza, 2014).
**3 Results and discussion**
**3.1 Dispersion correction and its effect on the input dataset**
The mean diel variations of the ventilation coefficient ratio and of its MLH and WS constituents are
shown in Fig. S1 separately for seasons. They all exhibited a pattern consisting of a broad band during
the daylight period. The MLH curves showed the maximum value in summer, a lower, but close time
series in spring, the minimum value in winter and a close, but somewhat higher curve in autumn (Fig.
S1a). During the evening and night, the curves were similar to each other. The WS time series displayed





maxima in spring, smaller and fairly similar levels in summer and winter, and minima in autumn (Fig.
S1b). As a result, the time series of the $VC_{ratio}$ over the peak region were similar to each other in summer,
spring and autumn, while the ratios in winter were smaller than for the other seasons (Fig. S1c). The order
of the levels of the VC time series during the evening and night were just the opposite to those over the
daylight. The $VC_{ratio}$ data were above unity (up to 2.5 in summer) approximately from 07:00 to 15:00
UTC+1 in all seasons, whereas they were < 1 (down to 0.25 in summer) outside this time interval. These
all indicate that the dispersion correction can be substantial in summer, spring and autumn, and it is
smaller, but still relevant in winter.

The effect of the dispersion correction on the PMF input data are demonstrated by the diel variations of
the uncorrected and dispersion-corrected $N_{6-1000}$ for separate seasons (Fig. 1). The structure of the
uncorrected curves (Fig. 1a) was discussed and explained earlier (Salma et al., 2011, 2020; Thén and
Salma, 2021). In summary, they show three peaks; early-morning peak and evening peaks at the rush
hours of 06:00−08:00 and 18:00−21:00, respectively, largely generated by vehicular road traffic, and a
midday peak primarily produced by NPF events driven by photochemistry. The concentrations from 23:00
to 05:00 monotonically decreased and were virtually identical to each other. The curve in summer seems
to be below the other lines during the daylight period.

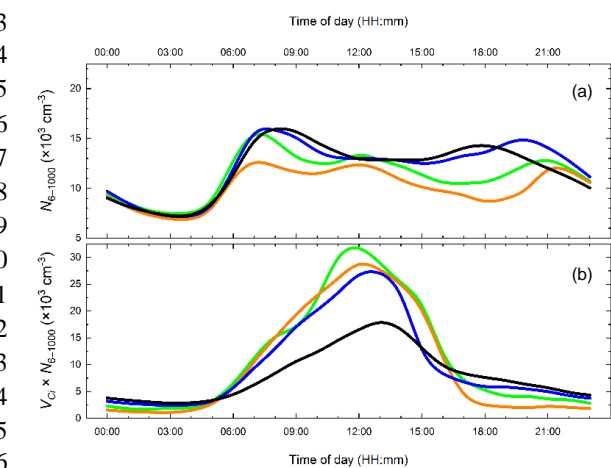

**Figure 1.** The mean diel variation of the uncorrected ($N_{6-1000}$; a) and dispersion-corrected total particle number concentrations
($VC_i \times N_{6-1000}$; b) separately for spring, summer, autumn and winter.

The concentrations and shape of the dispersion-corrected diel curves were vastly different from the
uncorrected lines (Fig. 1b). They all consisted of a broad, single structured peak. The largest maximum




of the peaks was observed in spring, the curves in summer and autumn were somewhat lower and similar
to each other, while the peak in winter was substantially lower than in the other seasons. The shift in the
timing of the maxima was influenced by the clock change for the daylight-saving periods. The curves
exhibited monotonically decreasing tendency in evening and reached a constant level during the night.
The concentrations of the corrected data during evening and night were smaller, while their levels during
the daylight period were larger than the uncorrected levels (as is expected from the $VC_{ratio}$ time series;
Fig. S1c). These results emphasize that the input data for the PMF modelling became different after the
dispersion-correction from the uncorrected dataset and better reflected the actual emission patterns.
**3.2 Interpretation of the factors**
The regression lines for the measured and uncorrected modelled $N_{6-1000}$ are show in Fig. S2. The curves
and their statistics indicate that the PMF modelling yielded reasonable agreement with the data. Based on
the selection criteria described in Sect. 2.2, six-factor solutions were accepted for both the uncorrected
and dispersion-corrected datasets and for each season. More factors resulted in unreasonable splitting of
some factors (even in winter), whereas a smaller number of factors yielded questionable merging the
factors. The approved final solutions represent physically meaningful and sensible approximation for
Budapest. The PMF results derived from the uncorrected input data are interpreted in Sects. 3.2.1–3.2.5.
The time tendencies and conditional bivariate probability plots of the outcomes obtained from the DC-
PMF modelling indicated qualitatively comparable properties and behaviours to them.
**3.2.1 Nucleation**
The factor associated with the smallest particles in our experimental setup was characterised by a single
mode in the source profile with a diameter range from 6 to 25 nm (Fig. 2a). This range ordinarily
represents the nucleation mode in NPF studies (Kerminen et al., 2018) and corresponds to its typical time-
averaged evolution (e.g., Salma and Németh, 2019). The factor contributions (concentrations) were the
largest in spring and the smallest in winter (Fig. 2b). This variation coincides with the relative occurrence
frequency of the NPF events in the larger Budapest area (the Carpathian Basin; Salma et al., 2016b, 2021).
The diel variations of the $N_{6-1000}$ from this factor showed the highest intensity at 12:00 in all seasons with
the largest peak in spring and with the smallest peak (if any) in winter (Fig. 2c).

Nevertheless, the time series unambiguously indicated additional peaks in the early-morning and evening
rush hours in addition to the midday peak (Figs. 2c and S3a, b). The factor also exhibited non-negligible
association with NO, NO$_2$ and CO with varying degrees (Fig. 2a). These results suggest that there is
connection between this factor and the vehicular road traffic, particularly in non-winter seasons. The



compound character of the factor was recognised earlier (Rivas et al., 2020). In our results, the importance
of the traffic-related sub-factor was higher on weekdays compared to weekends (particularly in the early-
morning rush hours on Sunday) when the traffic intensity is lower (Fig. S3a). The small peak at ca. 110
nm could be generated by heterogeneous nucleation of semi-volatile organic compounds on primary soot
particles, which is a likely process in rapidly diluting and cooling air due to the turbulence caused by road
vehicles. It could equally be a modelling artefact since in this diameter range, enlarged displacement
intervals were noticed.

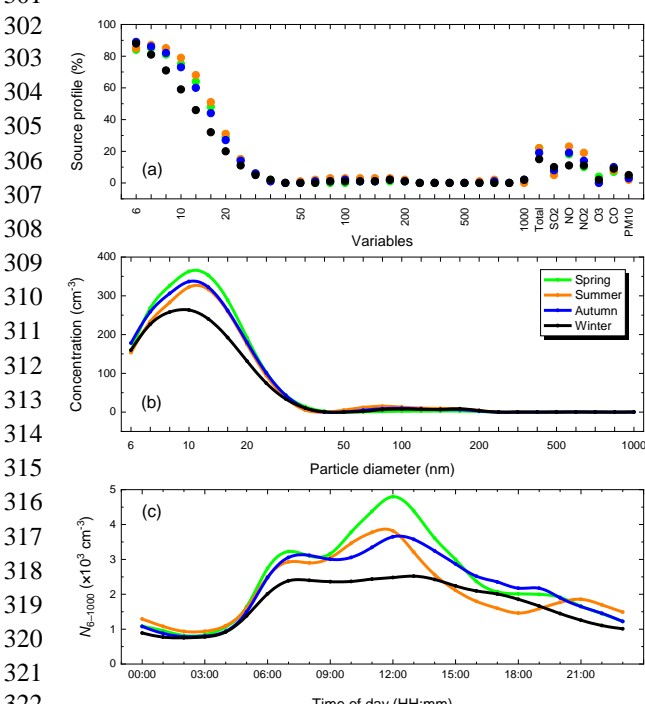

**Figure 2.** Relative factor profile (a), factor contribution to the particle number concentrations in the size channels (b), and the mean diel variation of the total particle number concentrations ($N_{6-1000}$; c) assigned to the compound nucleation source in spring, summer, autumn and winter. The exact diameters of the size channels are listed in Sect. 2.1.

This factor is interpreted as nucleation that is a combination of photochemically induced nucleation with
traffic-related nucleation. The former process occurs on a regional or urban spatial scale around noon. In
our results, this was also associated with strong southern winds (Fig. S9) consistently with our earlier
conclusions (Németh and Salma, 2014). Higher WS values often represent cleaner air in the city centre,
and the relationship between high WS and NPF occurrence is in line with our earlier observations in
Budapest (Salma et al., 2021). The traffic-related nucleation in cities can happen when the gas-phase
vapours and gases in the exhaust of road vehicles cool, and the resulted supersaturated vapours can





nucleate outside the source (Charron and Harrison, 2003). The process yields nucleated particles which have been called delayed primary particles (Rönkkö et al., 2017). This explains why the traffic circulation patterns showed up in the time series of this factor.

**3.2.2 Traffic emissions**

There were two factors showing unimodal source profile each in the Aitken mode, which indicates that these were primary particles (Figs. 3a and 4a). Both factors exhibited considerable contributions to NO, $NO_2$ and CO as well. These gases are related to combustion processes. The time series of the concentration contributions of the two factors clearly followed the daily and monthly patterns of the vehicle circulation in Budapest, and were larger on weekdays than on weekends (Figs. 3c, 4c, S4 and S5). They both can be related to direct emissions from motor vehicles. There were, however, several major differences between the two factors, which discriminate them from each other.

One of the road traffic emission factors showed the largest contributions to the particles with a diameter of 25–35 nm (Fig. 3a). Its concentration contributions resulted in a mode, which was the smallest in summer (Fig. 3b). The diel variability of the factor also showed different magnitudes over seasons. The seasons were characterised by diverse seasonal mean $T$ values from 3 to 23 °C (Table S2). The contributions to the total particles were the largest in winter, large in autumn and spring, and the smallest in summer (Fig. 3c). This points to the presence of chemical constituents with semi-volatile physicochemical properties. The curves for summer contained a midday peak in addition to the rush-hour peaks, which could be related to the altered traffic pattern (with a peak at noon) in Budapest on summer holidays. The source origin was shifted to more regional scales with WS in spring, and showed local origin in winter (Fig. S9).

Based on these reasons and consistently with earlier conclusions (Robinson et al., 2007; Morawska et al., 2008; Rönkkö et al., 2017; Harrison et al., 2018; Rowell et al., 2024), this factor is interpreted as emission source of semi-volatile aerosol fraction from road vehicle traffic (traffic-svf). Considering that diesel vehicles are responsible for much of the exhausted particle numbers from road traffic in Europe (Damayanti et al., 2023), the important concrete source is the semi-volatile emissions from diesel engines. Emissions from gasoline combustion in spark-ignited engines likely contribute as well, which can be inferred from the differences in the diel patterns of the two traffic-related emission sources over the week (Figs. S4a vs. S5a). The naming and detailed interpretation of this factor varies in the literature such as emissions from gasoline vehicles (Liu et al., 2014) or fresh traffic emissions (Rivas et al., 2020) or Traffic 1 (Hopke et al., 2022).




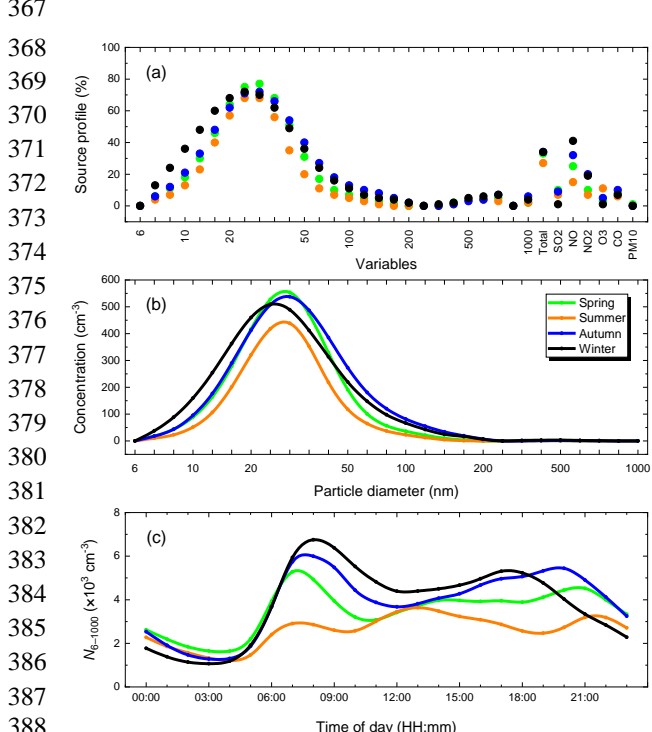

**Figure 3.** Relative factor profile (a), factor contribution to the particle number concentrations in the size channels (b), and the
mean diel variation of the total particle number ($N_{6-1000}$; c) assigned to the source of semi-volatile aerosol species emitted by
vehicle road traffic (traffic-svf) for spring, summer, autumn and winter. The exact diameters of the size channels are listed in
Sect. 2.1.

The other road traffic emission factor yielded a source profile in a broader diameter interval, actually with
a plateau over 65−140 nm, than the traffic-svf source (Fig. 4a). The factor also yielded higher
contributions to $SO_2$ and $PM_{10}$ mass. Its contributions to particle size channels exhibited a single mode
with a diameter of 90 nm, which were more stable over the seasons as far as the magnitude and shape are
concerned (Fig. 4b). The shares of this factor on the $N_{6-1000}$ did not seem to be influenced by the air
temperature in the seasons (Fig. 4c). The diel curves were shifted in the horizontal direction due to the
clock adjustments because of the daylight-saving periods. The source origin was related to smaller WS;
hence, it remained on more local spatial scale in spring and winter (Fig. S9).



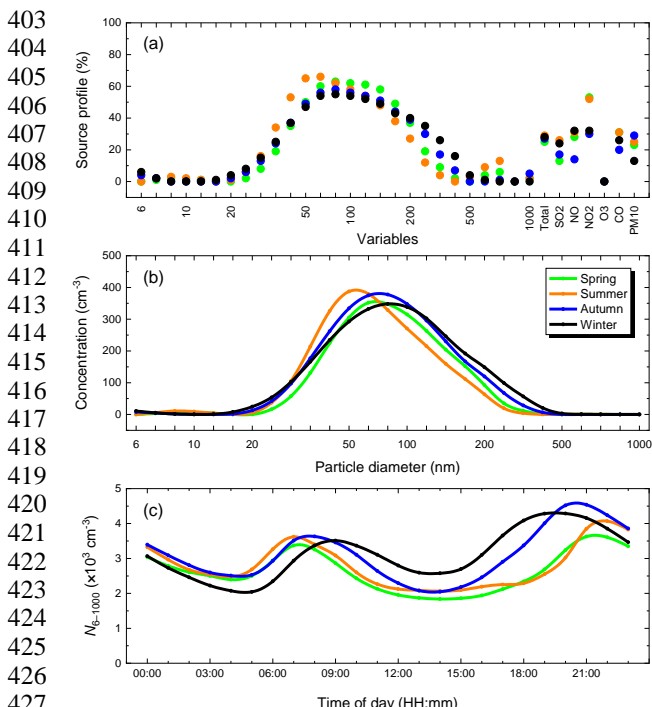

**Figure 4.** Relative factor profile (a), factor contribution to the particle number concentrations in the size channels (b), and the mean diel variation of the total particle number ($N_{6–1000}$; c) assigned to the source of solid aerosol species emitted by vehicle road traffic (traffic-sf) for spring, summer, autumn and winter. The exact diameters of the size channels are listed in Sect. 2.1.

Based on these reasons and consistently with the earlier studies (Maricq et al., 2002; Rönkkö et al., 2017; Damayanti et al., 2023; Rowell et al., 2024), this factor is interpreted as the source of solid aerosol species emitted by vehicle road traffic (traffic-sf). These particles likely consist of a soot core coated with varying amounts of low-volatility organics or inorganic compounds. The most important source contributing to this factor are the emissions from heavy- and light-duty vehicles (Zhang et al., 2020), which typically contain diesel-powered engine. Chemically and physically aged traffic particles can be partly involved as well (Robinson et al., 2007). The naming and the detailed interpretation of this factor varies in the literature, e.g., emissions from diesel vehicles (Ogulei et al., 2007) or Traffic 2 (Hopke et al., 2022).

### 3.2.3 Diffuse urban source

Another factor showed a profile with broad peaks at ca. 100 nm and 500 nm (Fig. S6a). It also contained several pollutants including $PM_{10}$ mass (typically in 30 % and up to 50 % in winter) and combustion-related pollutants such as CO, $SO_2$, NO and $NO_2$. The profile and contributions also included a low portion of smaller particles (around $d = 20$ nm). The contributions to concentrations exhibited structured multiple peaks between 70 and 500 nm, which showed elevated levels in winter and autumn, and low values in





summer and spring (Fig. S6b). The diel variations from spring to autumn displayed an early-morning
peak and an evening peak (with higher level in autumn and lower levels in spring and summer). This
pattern could be related to secondary particle formation from gas-phase precursors present in vehicle
exhaust when it is fully diluted within the ambient air and oxidised by reactive atmospheric species. In
such cases, the particles can grow by condensation. In winter, the diel variation was at the highest level
and was featureless (Fig. S6c). The factor was mainly linked to local spatial scales (Fig. S9).

Based on these considerations and earlier studies (Beddows et al., 2015; Beddows and Harrison, 2019;
Chandrasekaran et al., 2011; Vratolis et al., 2019; Wang et al., 2019b), this factor is interpreted as source
of diffuse (fugitive) urban aerosol. Important concrete sources contributing to it are aged combustion
emissions from various boilers and heating equipment used for residential heating or household cooking
activities. Burning residual oil and flaming combustion of solid fuels produce distributions with a modal
diameter at ca. 100 nm, while efficient combustion of gases and low viscosity oil in stationary burners
generate small particles (with a diameter around 20 nm; Hopke et al., 2022 and references therein). This
factor was called as urban background (Beddows and Harrison, 2019) or heating (Hopke et al., 2022).

**3.2.4 Secondary inorganic aerosol**

One of the further factors exhibited a source profile with a relatively narrow mode at the diameter of 800–
1000 nm and a broad mode from 50 to 150 nm (Fig. S7a). The larger mode was present in all seasons
with similar shapes to each other, but its concentration contributions were all negligible (Fig. S7b). The
smaller mode in the source profile was the largest in spring, smaller in summer and missing in autumn
and winter (Fig. S7a). Their concentration contributions in the size channels were modest. The shares
over a broad size range from 30 to 170 nm were relatively larger with a maximum of 120 cm$^{-3}$ in spring,
and with 70 cm$^{-3}$ in summer (Fig. S7b). These contributions were negligible in autumn and winter. An
addition mode in the contributions was observed at 250–400 nm, which seemed to be larger in winter
than in summer.

Based on these reasons and earlier conclusions (Squizzato et al., 2019; Hopke et al., 2022 and references
therein), this factor is ascribed to the sources of secondary inorganic aerosol (SIA), essentially of sulfate
and nitrate particles. An important concrete source types in our case could be their secondary formation
from gaseous precursors in motor vehicles exhaust (Yoshizumi, 1986). The sulfate particles are produced
preferably in summer and spring, when the photochemical activity is larger in a size mode around 100
nm (Yoshizumi, 1986). Consequently, their formation in winter is lower. The ammonium nitrate particles
behave contrary to this. They are mainly present in winter, when their thermal dissociation is low and in





a size mode at ca. 250 nm (Kadowaki, 1977; Squizzato et al., 2019). The seasonal tendencies and size
modes suggest that sulfate particles prevailed to nitrate particles in Budapest. The multimodal
directionality plots can indicate the presence of particles of both local and more distant origin. The latter
particles were likely influenced by gas-to-particle conversion or other atmospheric or cloud processing
(Ogulei et al., 2007; Kasumba et al., 2009; Squizzato et al., 2019).

**3.2.5 Secondary aerosol associated with high-ozone conditions**

There was a factor associated with remarkably high $O_3$ ($> 80$ %) and high $SO_2$ (40–60 %) contents. It also
showed a major mode in the size channels at the diameters of ca. 200 nm in summer (Fig. S8a). The
corresponding mode in spring was also present, but it became negligible in autumn and winter. This could
be caused by the large seasonal variability of $O_3$ in Budapest (Salma et al., 2020). As far as the factor
contributions are concerned, they exhibited a mode at ca. 45 nm in winter and autumn, and a different
mode at 150–200 nm in summer and spring (Fig. S8b). However, the absolute concentration contributions
to the size channels remained extremely low ($< 85$ cm$^{-3}$). These are in line with earlier studies, in which
a variety of size patterns with multiple modes were obtained (Ogulei et al., 2007; Liu et al., 2014;
Squizzato et al., 2019). The diel variation of the factor intensity during the daylight period in Budapest
was similar to the typical daily development of the in situ $O_3$ concentration in cities (Fig. S8c), and the
contributions were higher on weekdays compared to weekends. The directionality plots the factor
intensity indicated associations with higher WS (Fig. S9).

This factor cannot be strictly interpreted in a conclusive manner. It is thought to be the appearance of
particles from various primary origins that were grown by condensation of secondary vapours generated
by photochemical oxidation driven by $O_3$ (Juozaitis et al., 1996; Hopke et al., 2022). It is indirectly
inferred from the diel variations of the contributions to $N_{6–1000}$ in different seasons (Fig. S7c) and from
the size modes in the concentration contributions (Fig. S7b) that this source contains substantial fraction
of organic compounds. Additional input data on chemical composition would be advantageous to better
clarify this factor. This factor was called $O_3$-rich secondary aerosol in earlier studies (Ogulei et al., 2007;
Liu et al., 2014; Squizzato et al., 2019).

**3.3 Importance of sources**

The seasonal median uncorrected modelled concentrations of total particle number were 7.1, 6.8, 8.2 and
$7.8 \times 10^3$ cm$^{-3}$ from spring to winter, respectively. The mean source contribution fractions of the total
modelled concentrations derived by both the uncorrected and DC-PMF approaches are displayed in Fig.





5 for separate seasons. The relative contributions of unaccounted sources with respect to the measured
$N_{6-1000}$ were estimated to be ≤ 2 %.

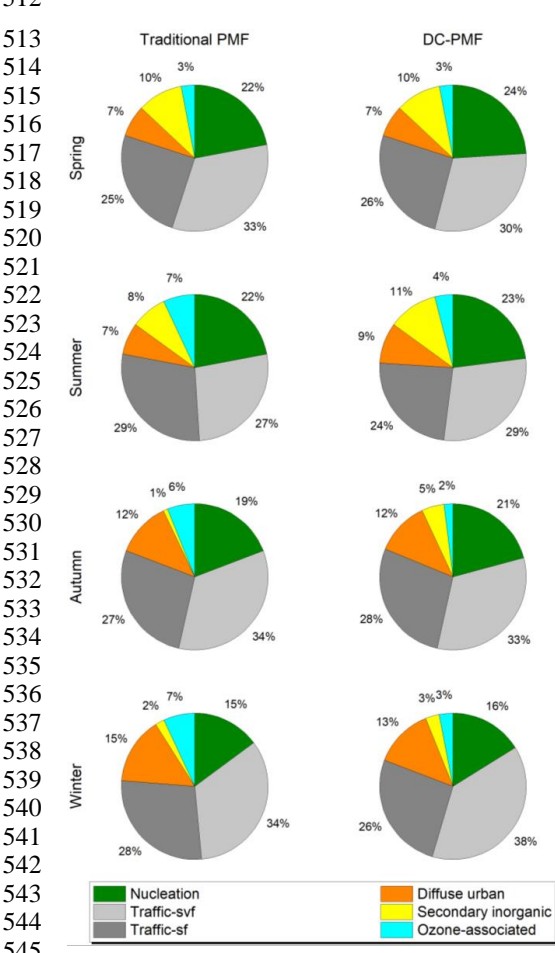

**Figure 5.** Relative contributions of nucleation, traffic semi-volatile fraction (traffic-svf), traffic solid fraction (traffic-sf),
diffuse urban, secondary inorganic aerosol and ozone-associated secondary aerosol sources to the modelled total particle
numbers as obtained by the uncorrected (traditional) PMF modelling and the dispersion-corrected (DC-)PMF modelling in
spring, summer, autumn and winter.

First we compared the effect of the dispersion correction on the source contributions. The correction
substantially enhanced the input concentrations from those sources that are typically active during the
daylight periods, and considerably reduced those that originate from the sources mainly active during the
nights. At the same time, the differences in the corrected-to-uncorrected ratios for the corresponding
contributions remained within 5 % in our datasets for the sources, which shared > 10 % of the $N_{6-1000}$.



Larger differences were only observed for the lower contributions, which raises the question of
interpreting the ratios obtained from small absolute values, and may indicate greater uncertainty in these
low values.

The overall mean relative contribution of the road traffic emission sources was 59 % (32 % for traffic-svf
and 27 % for traffic-sf). They did not show clear trend in seasonal variability. The values and properties
are in line with those in other European cities (Beddows et al., 2015; Brines et al., 2015; Dall'Osto et al.,
2012; Liu et al., 2014; Posner and Pandis, 2015; Squizzato et al., 2019; Rivas et al., 2020, Hopke et al.,
2022 and references therein). Despite that the emissions from vehicles can depend on multiple conditions,
for instance on the car fleet, general technical conditions of vehicles, properties of fuels and lubricants
used, driving conditions and even on the distance to the nearest road (Rönkkö et al., 2017).

The nucleation source was responsible for 20 % of the particle numbers annually. It was smaller in winter
than in the other seasons, particularly compared to the spring and summer. Its share was comparable to
our earlier conclusion of 12–27 % (to UF particles) as a lower assessment provided by nucleation strength
factor, and to indirect indications (Salma et al., 2017; Thén and Salma, 2022). The present contribution
of the nucleation can be, however, considered again as a lower estimate since an extensive portion of the
other sources, particularly the SIA in summer and spring and possibly also the urban diffuse source in
winter and autumn can be also related to the nucleation. The former source could partly contribute to the
nucleation through the vapours generated from gaseous precursors (including $SO_2$, $H_2SO_4$ and volatile
organics) in the exhausts of road vehicles, ships or airplanes and in the fumes of coal-fired power plants.
The urban diffuse source could be linked to nucleated particles via particle growth followed by physical
and chemical ageing processes, and possibly coagulation. An unusual type of nucleation events induced
by some urban, industrial or leisure activities on sublocal spatial scales with extremely high formation
rates was observed in Budapest several times (Salma and Németh, 2019). The contributions from the SIA
and urban diffuse source types were approximately 10 % in spring and summer, and 12−15 % in autumn
and winter, respectively. They could noticeably further enhance the importance of the nucleation source.
The $O_3$-associated secondary aerosol made up the smallest (6 %) mean contribution on an annual time
scale. The shares of the SIA in winter and autumn were 2−3 %. These tendencies are in line with our
general understanding of the time behaviour of the related sources and particles.

The directionality plots for the uncorrected PMF results for separate sources are presented in the first two
columns of Fig. S9 for the most informative season pairs (for which differences in the $N_{6–1000}$ were the





most noticeable). The road traffic emission sources were related to local spatial scales in all seasons except
for summer. In this latter case, more distant regions and larger WS values prevailed. The nucleation source
in spring (when its occurrence frequency was the largest) was associated with SE direction and high WS.
This directionality is coherent with our earlier finding (Németh and Salma, 2014). In winter, its source
directionality plot was featureless. The diffuse urban aerosol originated from local spatial scales and low
WS in all seasons, which is in accordance with its source interpretation. The SIA was relevant only in
spring and summer, with prevailing SE and NW directions, respectively and with high WS values. The
intensity of the $O_3$-associated secondary aerosol source in winter and autumn remained low in the city
centre and higher in its outskirt.

The differences in the directionality plots were obtained by subtracting the uncorrected PMF results from
the DC-PMF results. They are shown in the third and fourth columns of Fig. S9 for the identical seasonal
pair as for the directionality plots. Despite the similar seasonal mean contributions from both the
uncorrected and corrected PMF (Fig. 5), there are substantial variations in the plots. The corrected PMF
can considerably change the source origins. In this respect, the DC-PMF can provide important added
values for interpreting the spatial distribution of the sources. More detailed and reliable interpretations
will be feasible after gaining further experience and expertise in the future studies.
**4 Conclusions**
Six major source types of particle numbers were identified in Budapest. The road vehicle emissions were
the largest contributors; they were responsible for approximately 60 % of particles. This source was
resolved into a semi-volatile fraction and a solid (soot core) fraction. It seems likely that these two types
do not express the emissions from gasoline- and diesel-driven motor vehicles, but they represent two
distinct groups of chemical mixtures from both internal combustion engines. Nevertheless, both sources,
particularly that containing solid fraction, are dominated by diesel motor vehicles. More importantly, the
latter source is characterised by a modal diameter around 90 nm and is expected to contain high portions
of insoluble particles. These properties can yield considerably larger lung deposited surface areas than for
the traffic-svf or the other sources (except for the urban diffuse source), which results in extraordinary
particle burden in the human lung caused by this individual source. Moreover, the surface-active
properties of soot core likely represent additional risk for the health outcomes.

The nucleation source was responsible for ca. 20 % of particles as a lower estimate. It displayed a
compound character consisting of photochemically induced nucleation and traffic-related nucleation.



There is a method available for splitting it into the two specific (sub)sources using $NO_x$ as a proximity
marker for vehicle road traffic (Rivas et al., 2020). However, in our datasets the coefficients of correlation
between the nucleation intensity and $NO_x$ concentration were typically $< 0.2$, and adopting this method
yielded unusually small photochemically induced nucleation contributions. They are in contrast with our
earlier results and other indirect estimations (Thén and Salma, 2022), and with other suggestions as well
(Rowell et al., 2024). Therefore, we avoided adopting this estimation for the time of being, and emphasize
here the need for developing generally valid splitting methods, and testing them on a variety of datasets.

All particle number size distributions attributed to the sources together with their relevant conjugate size
distributions are to be further utilised in an advanced lung deposition model for characterising and
quantifying source specific depositions in the human respirators system.
*Data availability.* The observational data are available from the corresponding author (IS).
*Supplement.* The supplement related to this article is available online at: *to be completed*.
*Author contributions.* MV performed the data treatment and modelling, prepared the figures, participated in the interpretation
and writing the manuscript. PKH participated in the conceptualization the interpretation of the results and editing. IS provided
the dataset, conducted the conceptualization, participated in the interpretation and writing the manuscript. All coauthors
contributed to the discussion of the results and provided comments on the manuscript.
*Competing interests.* The authors declare that they have no conflict of interest.
*Financial support.* This research has been supported by the Hungarian Research, Development and Innovation Office (grant
K132254), and the New National Excellence Program of the Ministry for Culture and Innovation from the source of the
National Research, Development and Innovation Fund (ÚNKP-22-3).

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
