# Peer review of "Attribution of aerosol particle number size distributions to major sources using a 11-year-long urban dataset"

_EGUsphere, 2024_

## Author Comment (AC1)

**Response to Referee number 1**

27 March 2024

The authors would like to thank Referee no. 1 very much for her/his expert, detailed and valuable comments which allow us to further improve and clarify the MS. We have considered all recommendations and made the appropriate alterations. As a follow up of a comment (of Referee no. 2), we realised that the dispersion correction was based on the seasonal mean VC values, and not on the mean VC averaged for the overall dataset. The latter choice is in accordance with the original idea and further adaptations of the method. Therefore, we repeated the calculations with the appropriate correction factor. Some numbers and figures were slightly modifed because of this, while the tendencies remained the same. To completely fulfill another request, we inserted a new section 3.3 Relevance of the dispersion correction, and consequently extended the text by more detailed discussions, performed structural alterations or reorganisations at several places in the MS. We also adopted some additional smaller changes. Furthermore, we prepared in total 10 new compound figures for various purposes and inserted them into the Supplement. They all improved the quality of the revised material and further validated or documented the interpretations. Our specific responses to the comments are as follows, while most textual modifications amended to the MS can be traced in its marked-up version, which is available online.

**Specific comments**

*Line 69. Please define WS the first time that it is used.*

Response 1: The definition of the abbreviation WS was shifted from Line 129 to Line 69 of the original MS.

*Lines 87-96. Regarding the use an the performances of receptor models, I suggest to mention the work of Belis et al (Atmospheric environment: X 5, 100053, 2020).*

Response 2: The reference to this review-type paper was also added.

*Line 105. Better sources rather than source.*

Response 3: Adopted.

*Lines 110-117. If I have well understood the measurements were taken at the second site only during one year and not in parallel and a direct comparison of the two sites was not provided. In the time series analysed, the data have been put all together? This should be mentioned here.*

Response 4: This was specified in Lines 141–142 of the original MS. Nevertheless, we inserted two brief sentences already at this place to indicate these details as: "The exact timings of the measurement years are detailed in Table S1. The experimental data from the two measurement sites were merged into one dataset which was evaluated jointly."

*Lines 168-179. What values are used for A? In addition, the C3 value of 0.2 was used for each channel or only for the total concentrations (i.e. the sum of all channels)?*

Response 5: The fine tuning of $\alpha$ for particle number concentrations was performed by changing the multiplication factor $A$ from 0.1 to 5. The factor $C_3$ was used for the total particle number concentration and pollutants. With this realisation, we avoided tuning the parameters on the sample-by-sample basis. Since these two values control the deviation of the variables from zero and from the measured concentrations, respectively, it was necessary to change some other parameters as well to achieve good fits. The final parameters are summarised in Table S3. The missing details of this aspect were adopted in the MS, a new Table S3 containing the final values of the uncertainty parameters, and new composite Figs. S1−S4 on additional uncertainty analyses were inserted to the Supplement.

*It is interesting the comparison between the traditional PMF and that corrected with ventilation coefficient. I have not understood if the correction has been done at hourly level in the dataset. If yes, how calm of wind or absence of wind have been treated? In addition, the comments in lines 599-605 seems to be oriented in looking at the corrected PMF as more reliable, however, this is not demonstrated. Is there any reason to think that results with correction are more reliable than the traditional ones? If not, better to modify this sentence.*

Response 6: The ventilation correction was performed using the hourly mean datasets. The hourly mean WS values were obtained from the 10-min WS and WD data using vectorial averaging. The occurrence of the zero hourly-mean WS value was very low in the resulted dataset; the share of WS $< 0.1$ m s$^{-1}$ was 0.06 %. An explanatory sentence was added as: "The ventilation coefficient represents the maximum volume into which the particles undergo dilution after their release into or formation within the ambient air per unit time. The main purpose of this data treatment is to correct each concentration data to have the same VC as the mean VC over the whole, 11-year-long dataset. The latter quantity was 1768 m$^2$ s$^{-1}$ in our case.

It is challenging to directly prove the higher reliability of the DC-PMF results with respect to the uncorrected outcomes. We added the following two arguments: "It was demonstrated earlier (e.g., for Budapest lastly in Salma et al., 2020) that the local meteorological properties can influence the ambient atmospheric concentrations and size distributions in cities in a comparable extent than the changes in the source intensities (Li et al., 2023). The dispersion correction was dedicatedly introduced to remove a large part of the extra covariance between the variables, which is frequently or enduringly caused by the common effect of the meteorology on all concentrations. This basic motivation already implies that the corrected concentrations and concentration contributions are expected to be closer to reality and of higher reliability than their uncorrected counterparts. At the same time, the correction did not considerably alter the source profiles, temporal behaviours and patterns. Furthermore, some previous papers have also demonstrated the value of the dispersion correction in estimating the source contributions (e.g., Dai et al., 2020, 2021; Hopke et al., 2024)."

*Discussion of Figure 1. The midday peak seems actually to be present only during the warm seasons, rather than in every season as it seems to be mentioned here. Better to adda a legend on Fig. 1.*

Response 7: The midday peak in the $N_{6-1000}$ diel variation is visually more obvious in spring and summer than in the other seasons. Its interpretation was based on both our current and previous research work indicated by the cited references. They jointly prove the diel structure. The sentence under consideration was reformulated to indicate the basis of our statement more precisely as: "Conclusively, there are three peaks present with variable relative areas in the diel variations; namely an early-morning peak and an evening peak at the rush hours of 06:00−08:00 and 18:00−21:00, respectively, largely generated by vehicular road traffic, and a midday peak predominantly produced by NPF events driven by photochemistry." The requested legend was also added to Figure 1.

*Line 339. I would not say contributions. The factors are loaded with NOx meaning that there is an association among particles in these factors and gas but not a contribution. The same for line 396.*

Response 8: The sentences were reformulated to: "Both factors were strongly associated with NO, $NO_2$ and CO as well." and to "The factor was also considerably associated with $SO_2$ and $PM_{10}$ mass."

*Section 3.2.3. Is it possible that this source includes a contribution from resuspended dust, for example road dust resuspended by traffic? This may be possible for particles around 0.5 μm or more, see for example Conte et al. (Environmental Pollution 251, 830-838, 2019).*

Response 9: The road and soil dust resuspension by moving vehicles cannot be excluded. Nevertheless, its contribution to $N_{6-1000}$ for the urban diffuse source is expected to be limited in Budapest. The $N_{6-1000}$ for this source showed lower levels in summer and spring than in winter and autumn (Fig. S6c), while the former seasons are generally dryer than the latter periods. Hence, the expected change in the resuspension intensity does not show up in the concentrations. It is mentioned that scattering de-icing mixtures of sand and salts on roads in winter, which could confuse this conclusion, is limited both in its extent and frequency in Budapest. A brief sentence was added: "In principle, resuspension of road and soil dust particles could also add (Conte et al., 2019) as a minor contributor in Budapest."

Imre Salma
for the coauthors

---

## Author Comment (AC2)

**Response to Referee number 2**

27 March 2024

The authors would like to thank Referee no. 2 very much for her/his expert, detailed and valuable comments which allow us to further improve and clarify the MS. We have considered all recommendations and made the appropriate alterations. As a follow up of the comment no. 9, we realised that the dispersion correction was based on the seasonal mean VC values, and not on the mean VC averaged for overall dataset. The latter choice is in accordance with the original idea and further adaptations of the method. Therefore, we repeated the calculations with the appropriate correction factor. Some numbers and figures were slightly modifed because of this, while the tendencies remained the same. To completely fulfill another request, we inserted a new section 3.3 Relevance of the dispersion correction, and consequently extended the text by more detailed discussions, performed structural alterations or reorganisations at several places in the MS. We also adopted some smaller additional changes. Furthermore, we prepared in total 10 new compound figures for various purposes and inserted them into the Supplement. They all improved the quality of the revised material and further validated or documented the interpretations. Our specific responses are as follows, while most textual modifications amended to the MS can be traced in its marked-up version, which is available online.

**Specific comments**

*The Abstract and the first section "Introduction and objectives" should emphasize more the novelty of this work and its importance for the scientific community. In particular, the abstract gives a simple summary of the results without highlighting their relevance and impact.*

Response 1: We emphasised the novelty of the overall study, and paid more attention to its relevance for the research community in the Abstract and Introduction and objectives as "The combined application of the size segregated particle number concentrations, wide range of the size channels, considerably long dataset, dispersion correction and modelling over separate seasons can lead to novel insights into the aerosol sources, transformation and transport processes of particle numbers in cities. Our conclusions can also contribute to the general understanding of the sources, transformation and transport processes of particle numbers in cities and to developing novel innovative air quality regulatory policy for the particle numbers." The Abstract was also shortened to put more emphasis on the main messages.

*Throughout the text, the authors put a lot of emphasis on the importance of correcting data for the ventilation coefficient to take into account atmospheric dilution also in the model output, which is of great interest. However, the results focus very little on this aspect. The authors merely discuss the DC-PMF results only in a few lines in Section 3.3 (lines 551-558 and 599-605), being very generic on the findings. For example, it is not very clear if (and eventually how) the dispersion correction altered the diel patterns of the sources and if the correction effects were more visible on specific sources. I would suggest expanding more the discussion of the DC-PMF and adding some results in the Supplementary Material (e.g., the equivalent plots of Figures 2-4 and S6-S8 for the DC-PMF factors).*

Response 2: The discussion of the similarities and differences between the uncorrected and DC-PMF results was considerably extended both in the MS and Supplement. We organised these aspects into a separate section 3.4 Relevance of the dispersion correction, and prepared and discussed 6 new composite plots showing the effects of the dispersion correction on the source profiles, concentrations contributions and diel patterns (Figs. S13–S18).

*Line 34: the meaning of "criteria" in this sentence is not very clear, please rephrase.*

Response 3: The "Criteria Air Pollutants" is a reserved expression related to a set of six air pollutants ($O_3$, PM, CO, Pb, $SO_2$ and $NO_2$) introduced by the US EPA Clean Air Act. Its initials were turn to capital letters to indicate this link better.

*Lines 42-43: this sentence is not very clear. Do the authors mean that inhalation of small insoluble particles can lead to increased health risk compared to the one related to coarse or fine particles having similar chemical composition? If yes, please rephrase this sentence.*

Response 4: The sentence was reformulated to clarify its meaning as: "Inhalation of very small insoluble particles can lead to excess health risk relative to the effects of the coarse or fine particles having similar chemical composition (Oberdörster et al., 2005; HEI Review Panel, 2013)."

*Line 55: "particles are usually emitted into the air": do the authors mean that these particles are typically emitted as primary aerosol? If yes, please specify.*

Response 5: Primary particles are emitted into the air, while the secondary particles are formed in the atmosphere. The emitted particles and the primary particles are largely synonyms.

*Line 62: particle number concentrations and size distributions cannot be considered as "pollutants" (also because gases are included in primary pollutants). I would suggest modifying this sentence as "Primary pollutants (including particle number concentrations and size distributions of primary particles)…".*

Response 6: The sentence was modified and shortened; the suggestion was virtually adopted.

*Line 73-75: This sentence is long and not fluent; please, rephrase it.*

Response 7: The sentence was changed to: "The shape of PNSDs is influenced by the formation and transformation processes of particles, and by meteorological conditions (Li et al., 2023)."

*Lines 147-150: I did not understand if the subsets on which the PMF was run included all seasons for the 11 years or if a single PMF run was performed on each season of each year. Can the author provide more details on how many subsets the PMF was run?*

Response 8: A clarifying sentence was added as: "The PMF modelling was performed separately on each season joined over 11 years."

*Lines 187-188: the authors chose a seasonal value of VC. Can the author better specify what they mean? Is it the mean over all the years or a different mean VC value was calculated for each season and each year? This question is connected to my previous doubts related to lines 147-150.*

Response 9: The $\overline{VC}$ value and consequently the related calculations were repeated considering the entire measurement interval of 11 years, in accordance with the original idea and latter adaptations of the correction method. The main purpose of this data treatment is to correct each concentration data to have the same VC as the mean VC of the whole dataset, thus over 11 years. The latter quantity was 1768 $m^2\ s^{-1}$ in our case. The part under consideration was modified, extended and clarified from several aspects.

*Line 198-199: what are the final values chosen for the uncertainty parameters?*

Response 10: The missing details were specified in the MS, and a new Table S3 containing the final values of the uncertainty parameters were inserted to the Supplement.

*Lines 202-203: I would suggest inserting summary results of the bootstrap and displacement analysis in the Supplementary Material, or at least comment a little bit on the results of these analyses.*

The requested additional results of the bootstrap and displacement analyses in form of four new composite figures (Figs. S1–S4) were inserted to the Supplement and were discussed in the MS.

*Figures 2, 3, and 4: for sake of clarity, it would be very useful to highlight inside these figures what factor they are referring to (e.g., adding the name of the factor as a title, in the legend, inside the plots or in the y-axis label). Moreover, I did not understand why the figures related to only the first three factors were reported in the manuscript and the remaining ones were displayed in the Supplementary Material. Of course, adding too many figures in the main text is not advisable, and I also think that putting all these details into a single figure would not be straightforward; maybe the authors can at least comment on why they gave more importance to the first three factors.*

Response 12: The names of the sources were indicated at the top of Figs. 2–4, S10–S12 and S13–S15 as titles to improve their fast identification. The MS contains 5 figures consisting of several (mostly 3) panels. We decided to display the compound figures for the first three source types with the largest contributions (Figs. 2–4) in the MS to formulate our primary messages in a focused manner and to avoid overcrowding. The other compound plots for the remaining three source types (Figs. S10–S12) were placed in the Supplement. The former sources represent together more than 80% of the particle number concentrations, and, therefore, they are of greater importance than the latter sources. The MS was extended by these aspects as: "The related plots for the three major sources are displayed in the article (Figs. 2–4), whereas those for the remaining three sources are shown in the Supplement (Figs. S10–S12) to communicate our primary messages in a focused manner."

*Line 487: "become negligible": If I understood correctly, I would not use the word "negligible", because the patterns in winter and summer are just slightly smaller than the spring one. I would rather say that the contributions are simply smaller.*

Response 13: Adopted as: "The corresponding mode in spring was also present, but its contributions in autumn and winter became smaller."

*Figure S9: I found this figure quite hard to understand. Firstly, I would recommend adding on the top of the plots aside to the season label also "uncorrected PMF" and "(DC-PMF-uncorrected PMF)". Secondly, I would also suggest adding the plots related to DC-PMF results, otherwise it is very difficult to figure out their features only just looking at the (DC-PMF-uncorrected PMF) differences.*

Response 14: The conditional bivariate probability plots obtained from both the uncorrected PMF and DC-PMF models indicated qualitatively comparable properties and behaviours to

each other. Adding two extra columns to this already very complex, but auxiliary figure would increase its unwanted over-sophisticated or complicated character. The extra titles of the columns were readily inserted.

**Technical corrections**

*Line 36: add a comma after "Despite that".*

Response 15: The sentence was modified.

*Line 103: I would suggest adding "atmospheric" before "dispersion correction" for sake of clarity.*

Response 16: The word was added.

*Line 157: I would suggest citing the original work by Paatero (1999) for the ME-2 solver (the reference already listed in lines 773-774).*

Response 17: The reference was cited.

*Line 168: add a space between "N" and "represents".*

Response 18: There is a space between the "*N*" and "represents", but due to the Italic style of "*N"*, it looks smaller than usual.

*Line 172-173: please provide references for this statement.*

Response 19: The following review-type references were cited: Hopke, 2020 and references therein.

*Line 182: "This effect can be corrected for by…": eliminate "for".*

Response 20: Eliminated.

*Line 223: add a comma between "winter" and "and".*

Response 21: Added.

Imre Salma
for the coauthors